# OpenReview forum: "Red Teaming Game: A Game-Theoretic Framework for Red Teaming Language Models"
_ICLR.cc/2024/Conference — ICLR 2024 Conference Withdrawn Submission_

### Official Review · Reviewer_f4uQ · 2023-10-30

**Soundness:** 2 fair
**Presentation:** 1 poor
**Contribution:** 2 fair
**Rating:** 3
**Confidence:** 4

**Summary:**

This paper presented a Red-teaming Game (RTG), a general game-theoretic framework without manual annotation, to study the alignment of LLMs. RTG is designed for analyzing the multi-turn attack and defense interactions between the Red-team language Models (RLMs) and Blue-team Language Models (BLM). They used the popular game solver, PSRO, to solve RGT. Empirical results in multi-turn attacks with RLMs show that GRTS autonomously discovered diverse attack strategies and effectively improved the security of LLMs, outperforming existing heuristic red-team designs.

**Strengths:**

This paper provided a game model to study the alignment problem of LLMs.
Through the Nash equilibrium, experiments show that the proposed approach improved the security of LLMs.

**Weaknesses:**

*It is not clear to me why RTG is a team game. For example, as shown in Figure 2, it is enough to model the game as an RLM and a BLM.

*This work assumes a team of RLMs, but some sentences are confusing: “It is worth noting that red team can be expanded to consist of multiple RLMs in RTG. In this work, we mainly discuss the red teaming task of single RLM.”

*This paper aimed to establish a rigorous mathematical model named RTG (Red Teaming Game) for the red teaming task of language models. However, the game is not formally defined. In the main text, the strategy space and utility function are not defined. Some symbols are not defined as well, e.g., \pi. The concept of Nash equilibrium is not formally defined.

*It is unclear how the proposed algorithm guarantees an \epsilon-approximate Nash equilibrium. That is, no theoretical results are provided.

*Without formally defining the game and the problem, it is hard to follow the experiment part.

*RLM is a bad guy, but this paper ‘aim to maximize the utility for RLMs and minimize the utility for BLM in multi-turn dailogue.’

Typo:
“illustrates The variation”
Caption (e) is missing in Figure 3
 “forms. topics”

**Questions:**

No

**Details Of Ethics Concerns:**

RLM is a bad guy, but this paper ‘aim to maximize the utility for RLMs and minimize the utility for BLM in multi-turn dailogue.’

---

> ### Author Response · Authors · 2023-11-21
>
> I acknowledge that presenting the Red-Teaming Game from a novel optimization perspective may appear intricate and challenging to those lacking fundamental knowledge in game theory. To cater to such readers, we have relegated a substantial amount of mathematical equations and definitions to the appendix, opting to retain only concise explanations in natural language within the main text.
> If your ethical concerns genuinely exist, does this suggest an acknowledgment of the efficacy of our approach in sufficiently optimizing the Red Team? This may seem inconsistent with your earlier comments and ratings; however, please be mindful that our methodology concurrently results in a more secure Blue Team. Lastly, "bad guy" may not be an appropriate term.

---

### Official Review · Reviewer_Kbyq · 2023-10-31

**Soundness:** 2 fair
**Presentation:** 1 poor
**Contribution:** 1 poor
**Rating:** 1
**Confidence:** 4

**Summary:**

This paper works to automate the process of finding and fixing problems with language models. The process is framed as a game between two teams. There is a red team which consists of a population of adversarial language models which take turns prompting the target model. The blue team consists of the target model which needs to be made robust to attacks. They iteratively sample dialogues, search for nash equilibria, and update the models accordingly. Beyond this summary, I do not understand many of the details of the paper. See below.

**Strengths:**

I like how the experiments are divided into many different categories of harm. I think it adds thoroughness and clarity.

**Weaknesses:**

1. “Existing work rely solely on manual red team designs and heuristic adversarial prompts for vulnerability detection and optimization.” is not true. There are many examples of automated black and white-box attack methods for LLMs.
2. The key to the paper’s contribution seems to be about using RL to train a population of adversarial LMs. I don’t see much value in how the paper wraps this process up in a game theoretic framework. Moreover, if a classifier for toxicity is already available, why not just perform adversarial training on examples that are selected by the classifier or optimized to elicit responses deemed toxic by it? These would seem to be strong baselines.
3. Relatedly, what is the purpose of using a population of red LLMs to each produce different turns in the conversation instead of a single one?
4. I find the presentation overall to be very poor and confusing. Much more than usual, I find myself struggling to understand parts of the paper after reading them multiple times. Algorithm 1 is an example of this. For example, it is not clear how the RLM and BLMs are initialized, whether exploitability is computed from individual turns or entire conversations, what the difference between the RLMs and policies are, what a “strategy” is (it is never defined at any point in the paper), what it means to compute a meta strategy (is this RL?), how a meta strategy can be computed insider the loop from U which seems to be initialized outside the loop and never updated, what the relationship between a strategy and a nash equilibrium is, what an oracle is, what a “missing entry” is, what a “diversity measure of semantic space” and what the star notation means. I simply do not think I can properly review this paper given how it is written.
5. There is no comparison to a baseline or benchmark.
6. Casper et al. (2023) is incorrectly described. It does not involve manual design of adversarial prompts. It involves using human feedback to develop a contextual measure of the harmfulness of text.

**Questions:**

Fig 3b. Why is reward not zero sum?

---

> ### Author Response · Authors · 2023-11-21
>
> Evidently, approaching the Red-teaming task from a multi-agent perspective is highly intuitive, allowing for simultaneous optimization of both the Red and Blue Teams. Additional motivation has been delineated in the abstract and introduction, obviating the need for redundant exposition here. To the best of our knowledge, no existing work presents a similar framework and solution, thereby precluding the establishment of a fair baseline for comparison. Due to constraints in length, a comprehensive detailing of all intricacies becomes unfeasible, posing challenges in accommodating readers lacking proficiency in game theory. However, it is imperative to note that pertinent literature references have been provided in the manuscript to guide interested readers in acquiring the necessary background knowledge. I express regret for any inconvenience, akin to the common understanding that explanations for why 1+1 equals 2 are typically unnecessary in most contexts.

---

### Official Review · Reviewer_qG6B · 2023-10-31

**Soundness:** 3 good
**Presentation:** 2 fair
**Contribution:** 3 good
**Rating:** 5
**Confidence:** 3

**Summary:**

This paper formularize the problem of red teaming as a multi-turn adversarial team game, where Red-team Language models (RLMs) play against a Blue-team Language model (BLM): RLMs aim to fool BLM into outputting harmful content, while the goal of BLM is to defend against this type of attack. The game is defined as a bi-level optimization framework $\mathcal G = (\mathcal T, \mathcal D)$. In the token-level $\mathcal T$, defined as a Markov decision process, a player aims to maximize the cumulative reward of sentences generated by its LLM. In the sentence level $\mathcal D$, the objective is to find an equilibrium strategy profile $\sigma^*$ of a multi-turn dialog game, formalized as an extensive form meta-game with payoffs defined by the returns obtained in $\mathcal T$. To solve the meta-game, the paper considers an approach based on two well-known methods, double oracle (DO) and PSRO. The paper conducts experiments using stablelm-alpaca-3b to evaluate the approach and to demonstrate its potential as a technique for ensuring alignment of LLMs.

**Strengths:**

Strengths:
- The paper is topical and interesting to read, but I have some concerns about the current presentation of the results, as written in the next comment section (weaknesses).
- To my understanding, game-theoretic approaches to red teaming / adversarial attacks have not been extensively studied in the literature on LLMs. Furthermore, much of the literature on red teaming and adversarial attacks focuses on "single turn" prompting, which doesn't capture sequential nature of dialog interactions. I think this is an important research direction, and the results of this paper provide an interesting starting point.
- The set of results suggest that the dialog meta-games, as formalized in this work, can improve the alignment of language models, making them less susceptible to jailbreaking attacks, and can be used as an oversight technique, indicating potential vulnerabilities or LLMs.

**Weaknesses:**

Weaknesses:
- As I mentioned in the previous section, the presentation of this paper could be improved. I find it strange that the formal setting is provided only in the appendix. Given that one of the main contributions of this work is a game theoretic framework, this formalism should be provided in the main part of the paper. Another important contributions, a measure of diversity in semantic space is also only explained in the appendix. At the moment, the main part of the paper doesn't look self-contained, and doesn't adequately explain relevant information. There are also quite a few typos in the paper, the font in some of the figures is too small, some of the figures take too much space (Fig 4a and 4d), and I generally believe that the clarify could be improved.
- The approach is grounded in the set of techniques previously considered in multi-agent reinforcement learning, but adapted to LLM dialog scenarios. From a technical point of view, the proposed approach appears to be a direct application/extensions of DO or PSRO to LLMs. That said, the application scenario is novel, and the paper considers a novel metric for semantic diversity.
- Generally, I found the set of experimental results somewhat limited, as explained below.
- The experiments focus on one model with 3B parameters: a) it's not clear whether the approach would scale to larger model sizes, given the complexity of the setting, b) it would be much more interesting to see whether weaker RLMs can perform red teaming of BLM that uses a different/larger LLM.
- The paper does not consider any baselines. Given that prior work has considered red-teaming with LLMs (e.g., (Perez et al., 2020)), one would expect some comparison to these approaches, e.g., in 1 turn scenarios.
- The paper limits the interaction scenario to 3 turns.  It's unclear why this hyper-parameter was set to this specific value. It would be useful to see if the performance of RLMs/BLM would degreade for larger number of turns.
- It's not clear why existing classifiers for hate speech were not used as an independent measure for toxicity. It's also not clear some other measures of diversity were not used in the evaluation, e.g., those based on Self-BLUE (e.g., as in Perez et al. 2020)).
- There are no ablation studies. The experimental results don't demonstrate the importance of having a population of RLMs.

**Questions:**

- The formal framework considers a red team with multiple RLMs, but the experiments seem to focus on only having 1 RLM. What is the benefit of having multiple RLMs? Could you also explain why Algorithm 1 is guaranteed to converge when we have multiple RLMs?

- A clarification question, in Section 4.3, it is written that the first round of attacks is composed of manually annotated adversarial prompts. Could you explain why manually annotated prompts were used instead of RLMs?

- In Fig. 3, some of the plots don't contain confidence intervals, e.g., Fig. 3a. Could you explain the rationale?

- Why do the experiments consider only 3 turns? How toxic are the outputs of the text generated by BLM? Did you consider any other metric for measuring diversity?

- How is ASR defined? How were the attack topics in Fig. 4 and 5 obtained?

- A somewhat orthogonal question: Have you tested the robustness of the trained BLMs against jailbreaking attacks (e.g., Zou et al., Universal and Transferable Adversarial Attacks on Aligned Language Models)?

---

### Official Review · Reviewer_HPCT · 2023-11-02

**Soundness:** 1 poor
**Presentation:** 1 poor
**Contribution:** 1 poor
**Rating:** 1
**Confidence:** 4

**Summary:**

Current red teaming requires human efforts to design prompts that trigger a toxic response from LM. This paper formulates the red teaming language model as a two-player game consisting of a red LM (RLM) and blue LM (BLM).

**Strengths:**

I appreciate the author's efforts in formulating red teaming from a game-theoretic perspective.

**Weaknesses:**

- Presentation is unnecessarily complex. Too many unnecessary acronyms harm readability. The value of formulating red-teaming as a game is unclear. In the introduction, the author said the drawback of prior works is the requirement of manual annotations, but the proposed method also requires manual annotation on the reward function. I agree the proposed method doesn't require humans to design the toxic or harmful prompts. But still, this doesn't directly lead to formulating red teaming as a game. I suggest the author focuses on explaining why game-theoretic formulation can mitigate the issue of red teaming instead of diving into a dense explanation of "how red teaming is formulated as a game." This is one of the major weaknesses of this paper that makes me give strong rejection.
- Presentation is poor. Figure 3 is extremely dense and can clearly be split into more figures. Also, the figures are barely visible.
- Diversity metric should be presented in the main paper, given that this is important in the proposed method. Similar for reward function definition, I found the definition of reward function in appendix but that is still far from clear. How you define reward is important in red teaming so I would suggest the author elaborate the definition of rewards.
- The presentation of experimental results in Section 4.1 is hard to follow. First, I cannot get the main idea and the purpose of this section in the first place. I can see the author is trying to explain how the evolution between RLM and BLM matches the formulation and arrives at Nash Equilibrium, but the presentation needs improvements. Also, the fact that BLM doesn't respond toxic text at prompts given by RLM doesn't imply that BLM won't generate toxic responses in other prompts. It's necessary to evaluate the safety of BLM in other prompts.
 - There are several diversity metrics in NLP. What's the difference between the author's diversity metric and NLP? The motivation for developing a new diversity metric needs to be justified.
-  How natural or fluent is the generated text? One concern when optimizing LLM with a reward model is that the RL policy can hack the reward model and generate unnatural text to get high rewards. Unnatural text are undesired in red teaming as red teaming is purposed for simulating interactions with human users, and humans won't generate that text. The requirement of generating natural text is the major distinction to adversarial attacks. I suggest the author conducts a human study to evaluate the naturalness of text. If the generated text are not natural, it would be a critical flaw that has to be fixed.
- Lack of baseline comparison. If the author wants to argue that the proposed method is a better automated red-teaming method, the author should compare with the existing red-teaming method to show the significance. For example, the baselines proposed in Perez et al. 2022 are valid baselines to compare with. This comparison is necessary to show the significance of this method.

**Questions:**

- Notations in Equation 1 is unexplained.

---

> ### Author Response · Authors · 2023-11-21
>
> First, I acknowledge that presenting the Red-Teaming Game from a novel optimization perspective may appear intricate and challenging to those lacking fundamental knowledge in game theory. To cater to such readers, we have relegated a substantial amount of mathematical equations and definations to the appendix, opting to retain only concise explanations in natural language within the main text. It is imperative to note, however, that the abstract and introduction sections have elucidated our motivation and core contributions. Through game-theoretic modeling, we have achieved simultaneous optimization for both the Red Team and the Blue Team, providing a theoretical framework for explicating the optimization process and objectives pertaining to language models. This establishes a fundamental paradigm for Red Team language model research, a facet hitherto absent in prior works, rendering a fair baseline for comparative analysis non-existent.
>
> Furthermore, any potential lack of familiarity with the aforementioned content may have deterred a more thorough examination of this manuscript, leading to less-informed critiques of aspects already presented in the text. We empathize with this circumstance but do not find it necessary to provide further elucidation in response.